# Study on Storm Surge Using Parametric Model with Geographical Characteristics

**Yeon-joong Kim** [1,*]**, Tea-woo Kim** [2] **and Jong-sung Yoon** [1,*]

1   Department of Civil and Urban Engineering, Inje University, Gimhae 50834, Korea
2   Department of Construction Technology Institute, Inje University, Gimhae 50834, Korea; burningwoo@inje.ac.kr
*   Correspondence: anyseason@inje.ac.kr (Y.-j.K.); civyunjs@inje.ac.kr (J.-s.Y.); Tel.: +80-55-320-3694 (Y.-j.K.); +80-55-320-3434 (J.-s.Y.)

**Abstract:** The coastal area of Japan has been damaged yearly by storm surges and flooding disasters in the past, including those associated with typhoons. In addition, the scale of damage is increasing rapidly due to the changing global climate and environment. As disasters due to storm surges become increasingly unpredictable, more measures should be taken to prevent serious damage and casualties. The Japanese government published a hazard map manual in 2015 and obligates the creation of a hazard map based on a parametric model as a measure to reduce high-scale storm surges. Parametric model (typhoon model) accounting for the topographical influences of the surroundings is essential for calculating the wind field of a typhoon. In particular, it is necessary to calculate the wind field using a parametric model in order to simulate a virtual typhoon (the largest typhoon) and to improve the reproducibility. Therefore, in this study, the aim was to establish a hazard map by assuming storm surges of the largest scale and to propose a parametric model that considers the changing shape of typhoons due to topography. The main objectives of this study were to analyze the characteristics of typhoons due to pass through Japan, to develop a parametric model using a combination of Holland's and Myers's models that is appropriate for the largest scale of typhoon, and to analyze the parameters of Holland's model using grid point values (GPVs). Finally, we aimed to propose a method that considers the changing shape of typhoons due to topography. The modeling outcomes of tide levels and storm surge heights show that the reproduced results obtained by the analysis method proposed in this study are more accurate than those obtained using GPVs. In addition, the reproducibility of the proposed model was evaluated showing the high and excellent reproducibility of storm surge height according to the geographic characteristics.

**Keywords:** storm surge; hazard map; typhoon model; GPV data; typhoon model

## 1. Introduction

Recently, the intensities of natural disasters have increased significantly owing to abnormal climate and various environmental factors; furthermore, natural disasters have occurred more frequently than before and have caused severe damages. In particular, Typhoon No. 21 "Jebi", in 2018, caused flood damage (Figure 1) [1] by a storm surge at the Kansai International Airport in Japan for the first time since its construction, in addition to severe property damage and casualties throughout Japan including Osaka. Hence, damage reduction measures against large-scale natural disasters are urgently required in this rapidly changing environment. The establishment of integrated measures is critical for mitigating the current natural disasters because they are characterized by their large scale, possibility of simultaneous occurrence of multiple disasters, and extremely high risk of danger (Kim et al. 2015) [2]. Furthermore, even if the cause of disaster is identical, disasters may appear differently depending on

the regional characteristics. Therefore, reduction measures should be established based on studies regarding disaster occurrence mechanism.

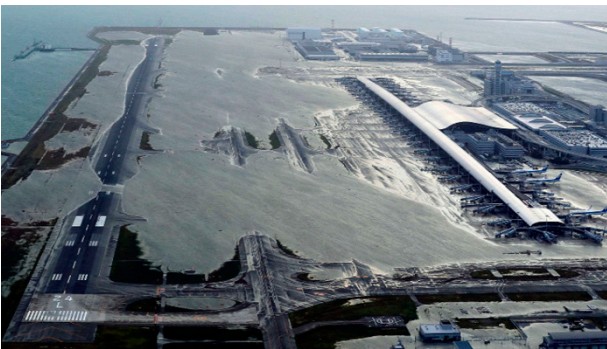

**Figure 1.** Flooding of Kansai International Airport in Japan caused by typhoon "Jebi" [1].

Damage cases from large-scale natural disasters are being reported worldwide. In particular, Japan has experienced the Great East Japan Earthquake in 2011 and is still striving to recover from the damage. Large-scale disasters can occur along with multiple disasters even by one single external force, and the damage can threaten the functions of cities and states. Consequently, based on experience gathered from large-scale disasters, the Japanese government is now establishing measures to prevent large-scale external forces and investigating the construction of disaster prevention systems in all disaster prevention fields. As a measure for large-scale storm surges, Japan has obligated the creation of hazard maps for storm surges by revising the Flood Control Act in 2015, published a hazard map manual (MLIT, 2015) [3], and conducted many reviews regarding frequent storm surge damage regions.

Many studies on storm surge have been conducted worldwide. Several researchers have shown that the coupling of surge, wave, and tide is a key element to improve the accuracy of total water level for coastal prediction [4–8]. Storm surge is a natural disaster that causes flooding in lowland and coastal areas due to a rise in sea level resulting from complex factors, including air pressure drop, wind, and high waves [9–13]. To estimate storm surges in Northeast Asia, the Myers model (Myers and Malkin, 1961) [14] is often used, which approximates pressure distribution near the center of a typhoon using concentric circles. Among the many causes of sea level rise due to storm surges, the rise in storm surge by wind changes significantly by the regional characteristics. Kim et al. (2007) [15] investigated the effects of land surface topography using the Mascon model to estimate sea surface wind in shallow waters. Chen et al. (2019) [16] evaluated the performance of SWH (significant wave heights) modeling for typhoons on the northeastern coast of Taiwan using different wind fields and a fully coupled tide–surge–wave model. Many researchers have also investigated wave transformation at the coastal area. In particular, Lee et al. (2017) [17] conducted numerical studies on the nonlinear interaction between the seawater and freshwater in a coastal aquifer and Kim et al. (2018) [18] analyzed the river discharge and salinity mixing due to wave on the geographical characteristics. Kim et al. (2019) [19] presented the changes in the sediment budget transport caused by long-term wave change. Moreover, Hsiao et al. (2019) [20] estimated the effects of wind sources from different spatial and temporal resolutions on storm wave height simulation and to investigate the optimal hybrid typhoon winds through two approaches for the best performance of storm wave height simulation.

The effect of large-scale storm surges was evaluated for the target region of this study, where the risk of flood damage is high owing to the large area of reclaimed land and lowland flats. In floods caused by storm surges, the duration of flooding is long, complicating disaster recovery and increasing secondary damage. In particular, the shape of a typhoon passing between Korea and Japan and moving to the East Sea is distorted from the shape of a concentric circle. Furthermore, in this region, the maximum sea level occurs several tens of hours after the closest typhoon approach.

In this study, among the typhoons that have occurred thus far, those with the possibility of the highest damage were classified by tracks based on the manual (MLIT, 2015) [3]. According to the

manual, storm surges are estimated by setting the central pressure and velocity of typhoons to the highest scale according to the moving tracks of past typhoons. The manual recommends the Myers (Myers and Malkin, 1961) [14] typhoon model. However, the Myers model, which is a concentric model, does not consider land topography when calculating wind and pressure fields [21–25]. The typhoon model for the target region must consider geographical characteristics to reproduce the maximum sea levels according to the geographical characteristics, and this study aims to propose a new analysis method that allows for topographic consideration. Consequently, the storm surge height reproducibility of the proposed method was evaluated to be excellent.

## 2. Analysis of Typhoon Characteristics

The 460 typhoons that landed or approached the target region were analyzed using digital typhoon data [26] from 1951 to 2016. The moving tracks of typhoons were classified according to the characteristics of the typhoons, as shown in Table 1. The sea level differences observed at the sea level monitoring station (Sakai) located closest to the target region were analyzed for each track. Subsequently, the typhoons in which the maximum sea level occurred are listed in Table 2. According to the analysis result, the maximum sea level occurred in Track 1, for which the typhoon moved to the East Sea after passing through Korea and Japan, as shown in Figure 2. In particular, the target region showed that the maximum sea level occurred approximately 10–30 h after the center pressure of the typhoon reached its maximum. Therefore, a storm surge model that considers the maximum sea level occurrence mechanism must be developed to investigate storm surges in the target region. The results of five selected typhoons and characteristics of sea level based on observed data at Sakai station are shown in Figure 3.

**Table 1.** Classification of typhoons by track.

| Track | Characteristics |
|---------|-----------------------------------------------|
| Track 1 | Track for moving between Korea and Japan |
| Track 2 | Track for landing in Japan |
| Track 3 | Track for bypassing China |
| Track 4 | Track for moving north along Japanese islands |

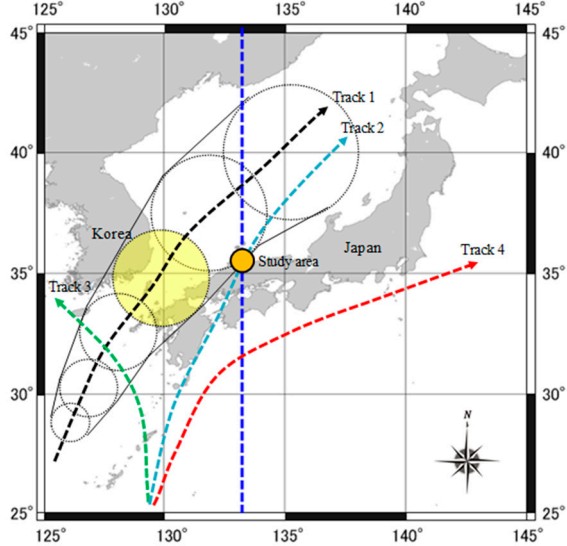

**Figure 2.** Classification of typhoons by Typhoon track at study area.

**Table 2.** The maximum tide and surge of the typhoon from the different track.

| Track | Number | Tide (m) | | Storm Surge (m) | |
|---|---|---|---|---|---|
| | | Max. | Top 5 (Avg.) | Max. | Top 5 (Avg.) |
| Track 1 | 133 | 1.102 | 1.041 | 0.630 | 0.560 |
| Track 2 | 159 | 0.962 | 0.856 | 0.400 | 0.358 |
| Track 3 | 22 | 0.776 | 0.763 | 0.250 | 0.227 |
| Track 4 | 146 | 1.002 | 0.836 | 0.554 | 0.365 |

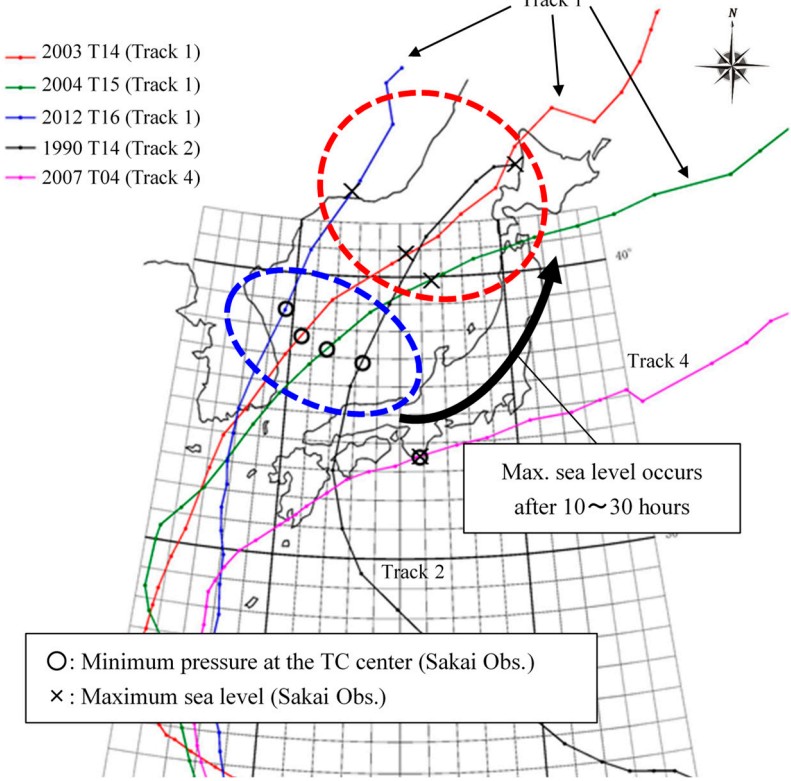

**Figure 3.** The results of five selected typhoons and characteristics of sea level based on observed data at Sakai station.

## 3. Building the Storm Surge Model

In this study, a storm surge prediction model was constructed based on the published manual (MLIT, 2015) [3].

### 3.1. Structure of the Storm Surge Model

The storm surge model is composed of a model for estimating pressure and wind fields (wind direction and velocity), a wave model, and a storm surge estimation model (Figure 4). The model was built considering the Coriolis force, air pressure variation, sea surface friction, and wave radiation stress using the Myers typhoon model; the SWAN model [27] for wave calculation; and the nonlinear longwave equation for storm surge estimation. The grid structure of the model is shown in Figure 5.

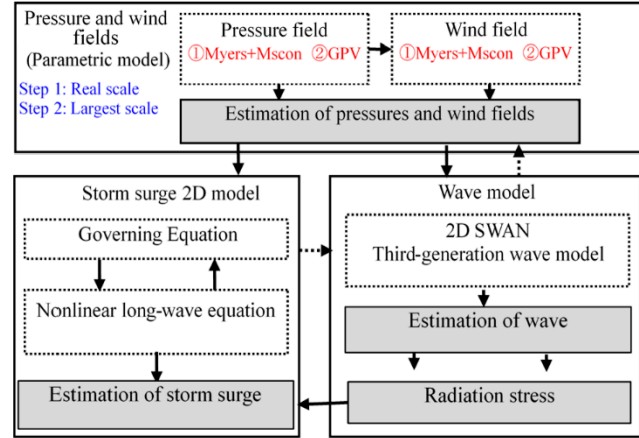

**Figure 4.** Foundation of storm surge models in this study.

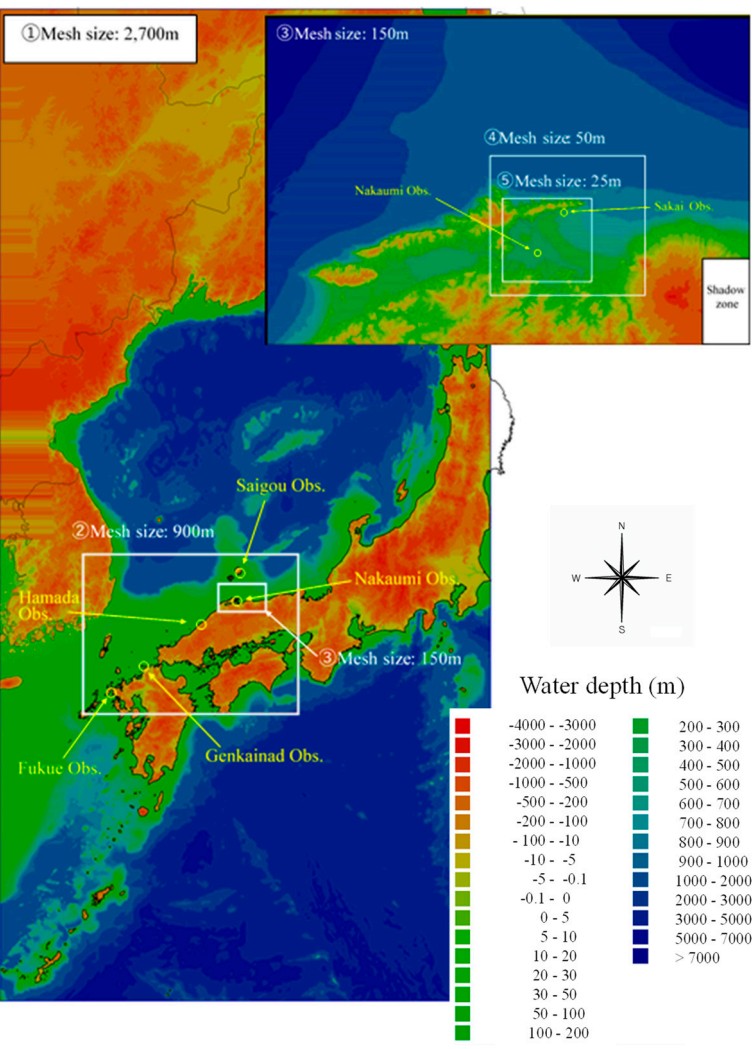

**Figure 5.** Computation domain with observation stations.

*3.2. Model Verification*

The proposed model was validated by estimating the pressure and wind velocity fields using the method suggested in the manual (Myers model) and by calculating the storm surge height using the grid point value (GPV) [28] analysis results provided by the Japan Meteorological Agency [29]. GPV data are from an archive maintained by the center for computational sciences, University of Tsukuba, for the

purpose of scientific development of the weather and climate forecasting technology. The Myers model calculates sea surface wind by assuming a concentric air pressure distribution and presents a limitation in reproducing wind and pressure fields, which change by geographical characteristics. However, the GPV data (MSM: grid size 5 km), which include the Japanese meteorological model used in this study, reproduce the wind and pressure fields well considering the effect of the wind field that varies by geographical characteristics.

*3.3. Results of Model Verification*

Figure 6 shows the reproduced and observation results using the Myers model and GPV data for the air pressure, wind velocity, wave height, sea level, and sea level differences of Typhoon No. 14 in 2003, which recorded the highest water level at sea level monitoring stations (Sakai and Nakaumi) near the target region. As shown in this result, the Myers model reproduces peak values of air pressure, wind velocity, and wave height relatively well. The observed value and model result of the maximum sea level difference were 0.63 and 0.60 m, respectively, indicating high reproducibility. However, the observed and estimated values of the maximum occurrence time were 9/13 19:00 and 9/13 5:00, respectively. Hence, the estimated value was reproduced 14 h earlier than the observed value. Meanwhile, the GPV model underestimated the peak value but correctly reproduced the peak occurrence time. This result validated the proposed model; however, the Myers model presented a limitation in estimating the storm surge height reflecting the geographical characteristics of the region.

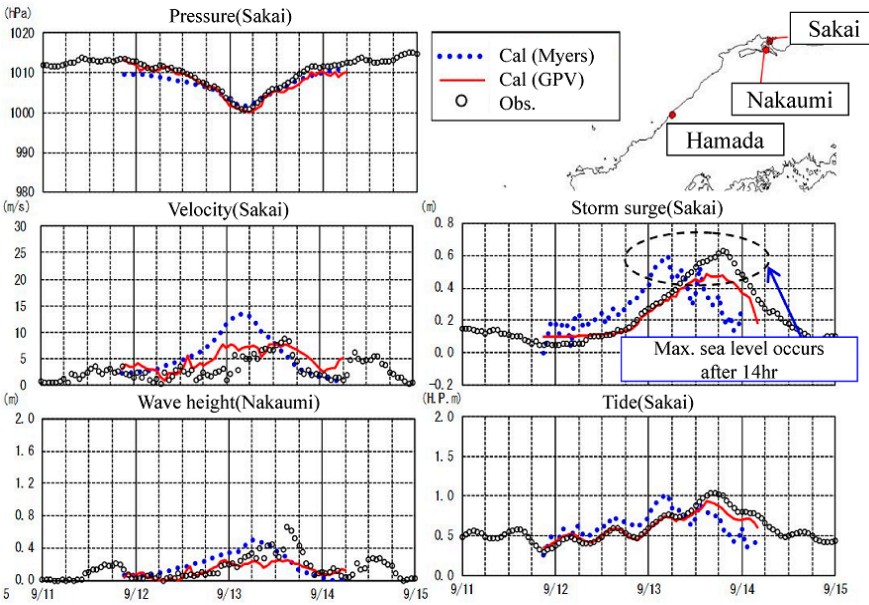

**Figure 6.** The results of the numerical simulation (Typhoon No. 14 in 2003).

Figure 7 shows the planar distributions of the wind direction and velocity using the Myers model and GPV data. A planar distribution was observed at 15:00 on 13 September 2003 when the peak wind velocity was recorded at Sakai station. This result confirms that the typhoon shape was altered by the geographical characteristics as the typhoon was moving to the north. In particular, the wind velocity at Point A in the region between Korea and Japan was less than 5 m/s for the Myers model assuming concentric circles, whereas it was higher than 10 m/s in the GPV data. This difference was presumed to be caused by the geographical characteristics of Korea and Japan while the typhoon was moving north toward the East Sea. Therefore, a new typhoon model must be developed because the Myers model assuming concentric circles failed to correctly reproduce the characteristics of the typhoon moving north toward the East Sea.

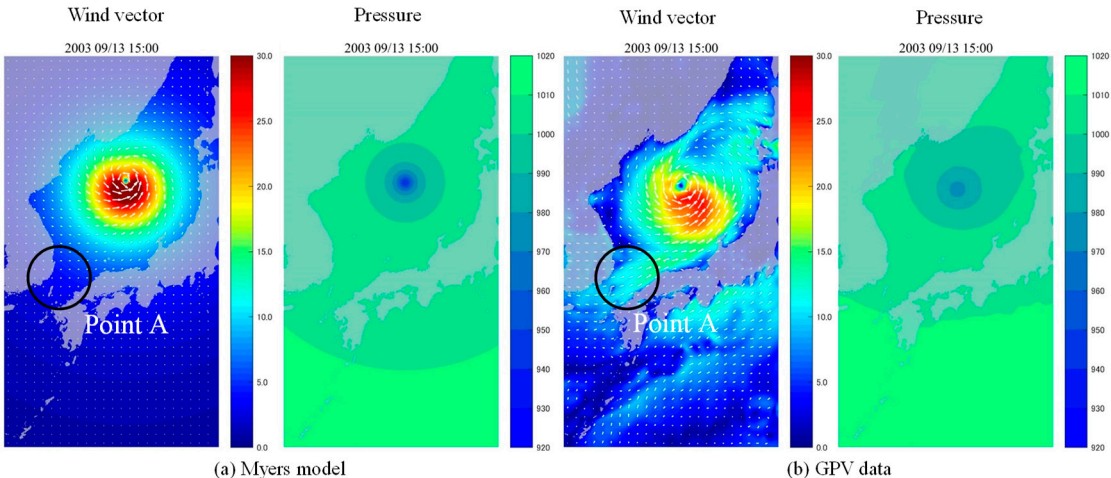

**Figure 7.** The results of wind vector and pressure based on typhoon models.

As shown in Figure 8, typhoons that develop in the ocean have a circular shape because of the lack of terrain influence there. However, as a typhoon approaches the shoreline, the surrounding terrain causes it to change from a concentric to an elliptical shape. This outcome indicates that, for the South Sea area facing the Pacific Ocean, it is possible to apply the Myers model assuming concentric circles but that, for the East Sea area, a model that takes into account the terrain influence is needed, because the typhoon shape is affected by this region's terrain.

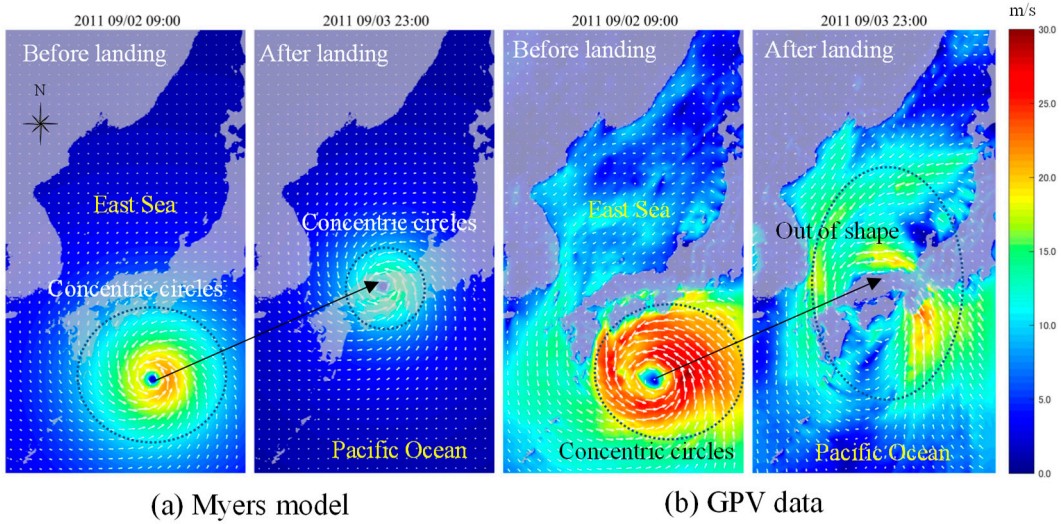

**Figure 8.** Typhoon shape changes based on terrain topography.

## 4. Typhoon Model

To reproduce the typhoon characteristics that generate the maximum sea level after the typhoon passes through the east coast, it is critical to reproduce the wind field that can consider the effects of land topography. The storm surge height caused by past typhoons can be estimated highly accurately using GPV data (since 2002). However, the ultimate goal of this study is to estimate the expected storm surge height of the highest scale, which requires examining the moving tracks of actual typhoons by setting the conditions of the largest typhoons according to the manual. Therefore, the construction of a new typhoon model that can consider topographical effects is required.

### 4.1. Analysis of Typhoon Shape Change

The locations of the center pressure and the air pressure distribution radii of the typhoons based on the GPV data are shown in Figure 9. When the typhoon passed through approximately 40° north latitude, the air pressure distribution radius (995 hPa: air pressure at the wind velocity (17 m/s) defined as a typhoon) and the effect range of the typhoon increased. This result shows that, as the typhoon moved north toward the East Sea, the typhoon shape changed from a concentric circle to an oval owing to the land topography.

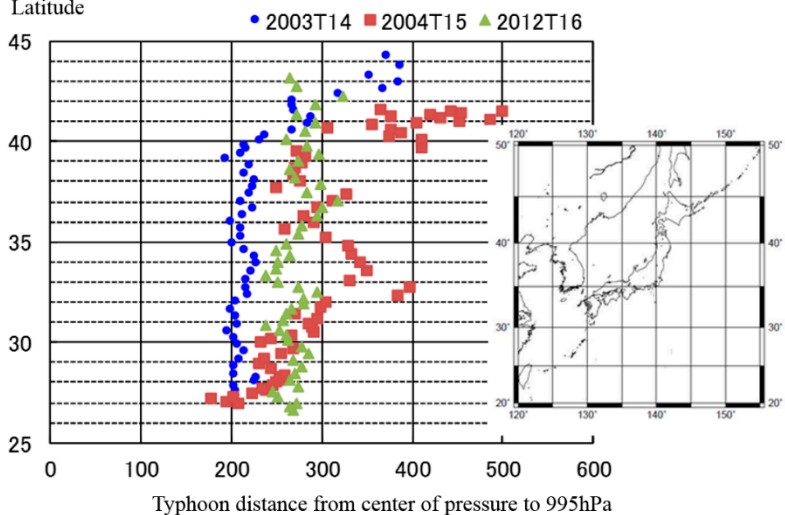

**Figure 9.** The results of wind direction based on typhoon models.

### 4.2. Holland Model

To reproduce a changing typhoon shape as it moves north toward the East Sea, the Holland model (Holland, 1980) (Equation (1)) [30] was adopted to reproduce the pressure field first, instead of the Myers model (Equation (2)), which is suggested in the manual. The Holland model [31] assumes the same concentric air pressure distribution as in the Myers model, but it can set the shape (gradient) of the air pressure distribution from the typhoon center to the periphery in more detail.

$$P(r) = P_c + \Delta P \cdot \exp\left(-\frac{1}{x^B}\right) \tag{1}$$

$$P(r) = P_c + \Delta P \cdot \exp\left(-\frac{1}{x}\right) \tag{2}$$

$$B = 1.5 + (980 - P_c)/120 \tag{3}$$

Here, $P(r)$ is the air pressure at $r$ km from the typhoon center, $P_c$ is the air pressure at the typhoon center (hPa), $\Delta P \ (= P_\infty - P_c)$ is the air pressure reduction, $P_\infty$ is the air pressure at $r = \infty$ (1013 hPa), $r_0$ is the distance from the typhoon center to the point of maximum wind velocity (km), x is $r/r_0$, and B is the gradient coefficient.

To calculate the gradient coefficient ($B$), which is a key variable of the Holland model, the gradient coefficient at which the reproduction error becomes the lowest was calculated using least-squares estimation for the GPV data, and the results are shown in Figures 10 and 11.

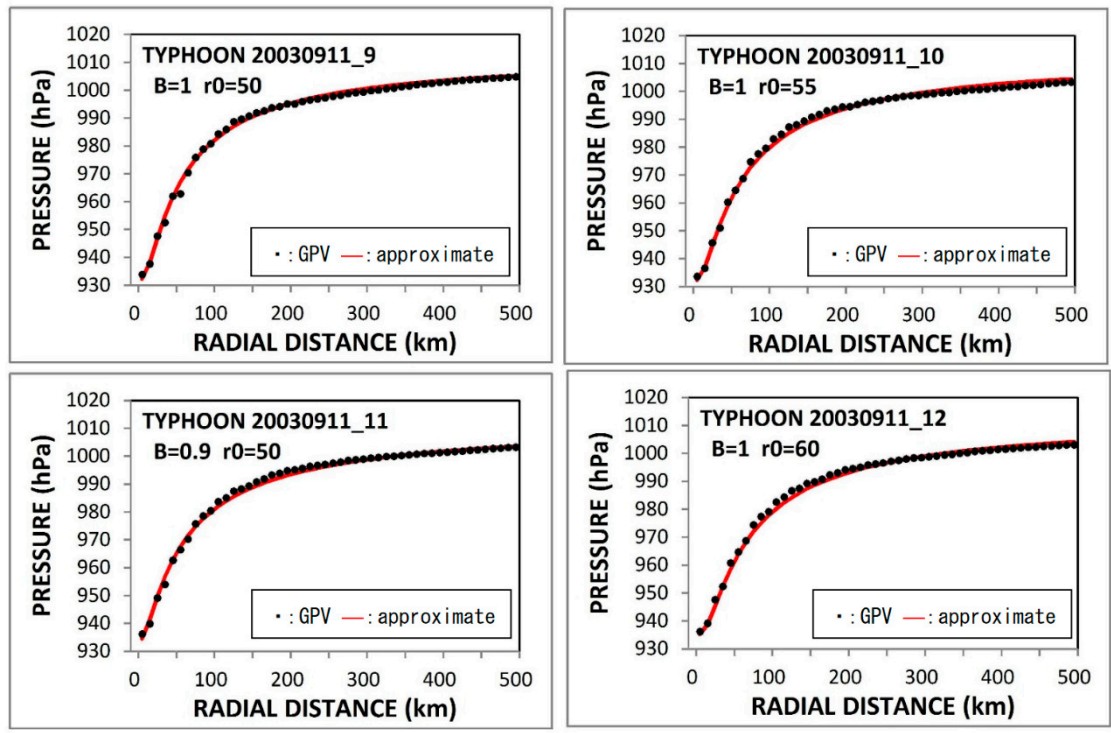

**Figure 10.** Variable calculation result of the typhoon model (Typhoon Maemi).

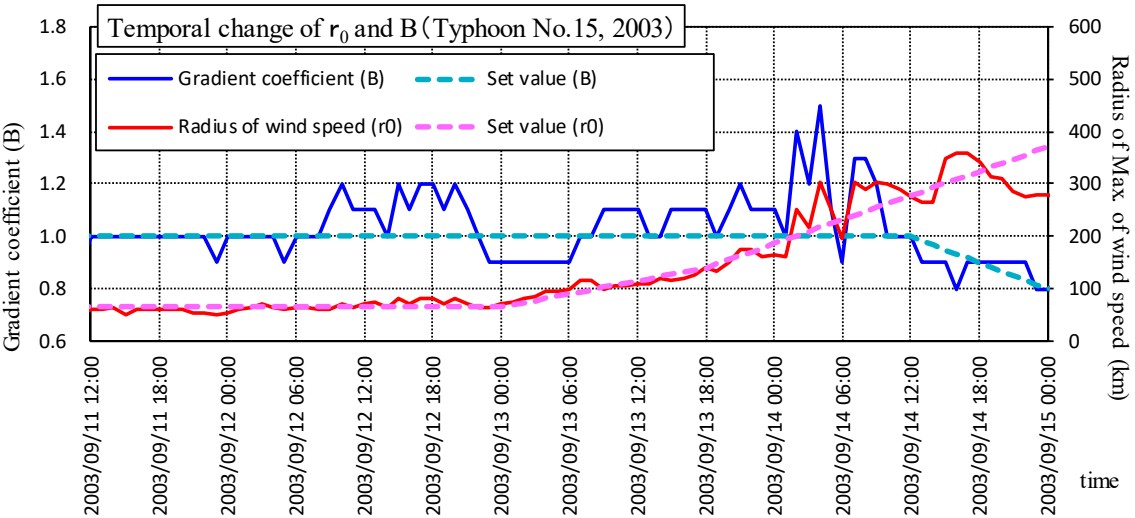

**Figure 11.** Temporal change of $r_o$ and $B$ (Typhoon Maemi).

### 4.3. Mascon Model

The Mascon model (Sherman, 1978) [32] recalculates the wind velocity to minimize the correction E of the variational equation (Equation (4)) based on the wind field obtained from the typhoon model.

$$E(u, v, w, r) = \int v \left[ a_1^2 (u - u_0)^2 + a_1^2 (v - v_0)^2 + a_1^2 (w - w_0)^2 + \gamma \left( \frac{\partial u}{\partial x} + \frac{\partial v}{\partial y} + \frac{\partial w}{\partial z} \right) \right] d_x d_y d_z \qquad (4)$$

Here, *u, v,* and *w* are the wind velocity components that satisfy the law of mass conservation; $u_0$, $v_0$ and $w_0$ are the initial wind velocity components; $\gamma$ is the Lagrange coefficient; and $a_1$ and $a_2$ are the variables related to air stability for the weight coefficient of the wind velocity correction.

### 4.4. Calculation of Wind Field Considering Geographical Characteristics

The Holland model was introduced to consider the typhoon model according to the characteristic of typhoons developed in deep sea and propagate to shallow sea, whereas the Mascon model was applied to reproduce the wind field that changes according to geographical characteristics. The application of the Mascon model in the wide area including this region is particularly important to reproduce strong winds that occur at Point A in Figure 7 after the typhoon has passed according to the maximum sea level occurrence mechanism due to geographical characteristics. Therefore, the wind and pressure fields were calculated using the Holland model for the entire region first, and the wind field was calculated using the wind field result value of the Holland model as an input of the Mascon model in Mesh 1 (mesh size: 2700 m) to evaluate the wind field that is changed by the terrain (Figure 12). Using these two results, the wind field change rate due to the topography in Mesh I was calculated. The wind field considering the geographical characteristics was calculated by applying the change rate for each region that was changed by the topography in the detailed regions (Meshes 2-5). The estimation flowchart is shown in Figure 13.

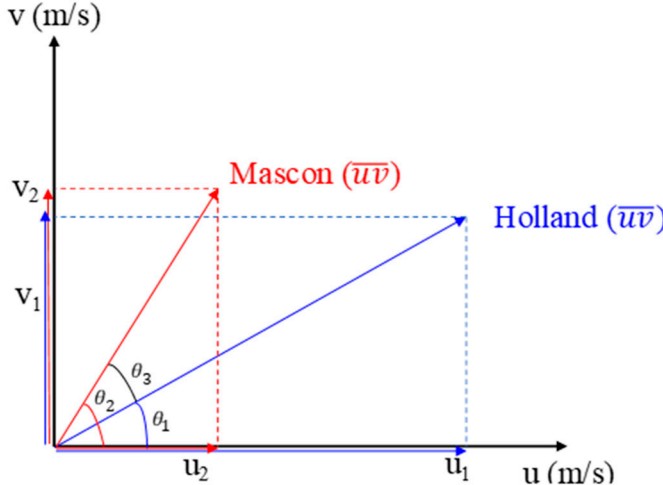

**Figure 12.** Wind velocity change rate in the topography of mesh (2700 m).

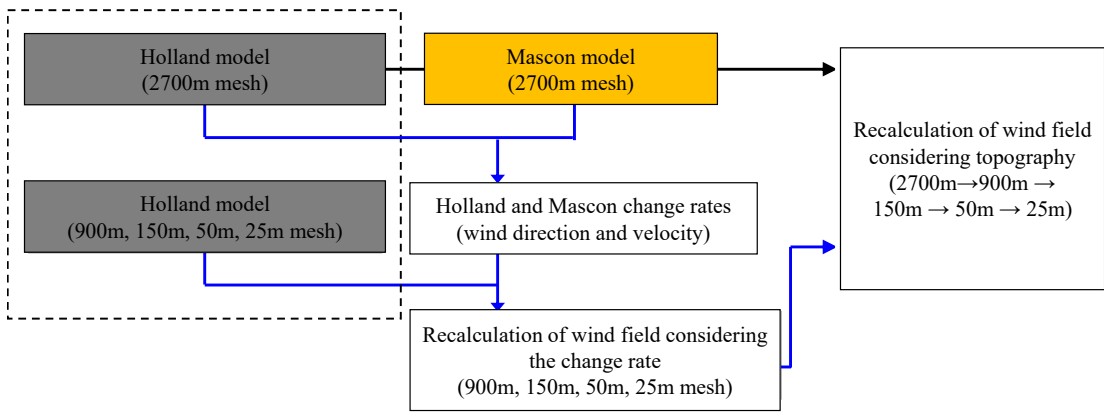

**Figure 13.** Wind field estimation flowchart considering topography.

Figure 14 shows the wind field estimation results applying the change rate altered by topography. As is shown, strong winds above 10 m/s at Point A (Figure 7) were not reproduced by the Holland model, but the Holland + Mascon models reproduced strong winds above 10 m/s, which were similar to the result of the GPV data.

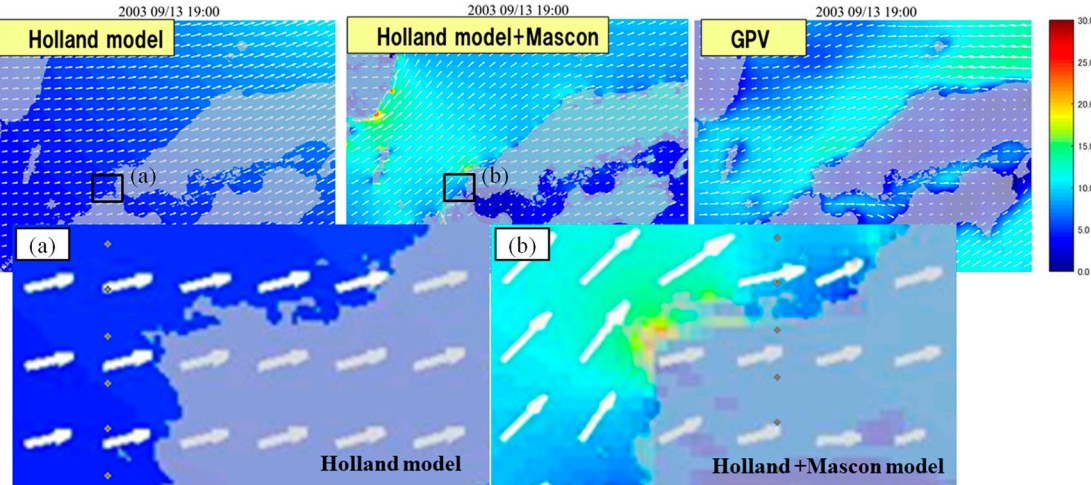

**Figure 14.** The results of wind velocity field considering topography (mesh: 900 m).

*4.5. Storm Surge Height Calculation Result*

To create the storm surge flood prediction map, a highly accurate reproducibility of the maximum sea level is required. However, in terms of disaster prevention, an accurate reproduction of the temporal storm surge height is crucial for flood control activities.

The estimation results of the wind velocity and storm surge height using the Myers model, Holland model + Mascon model, and GPV data at the Sakai station are shown in Figure 15. The reproduced values using the analysis method proposed in this study are similar to the estimation result using the GPV data. These results indicate the extremely high reproducibility of the model.

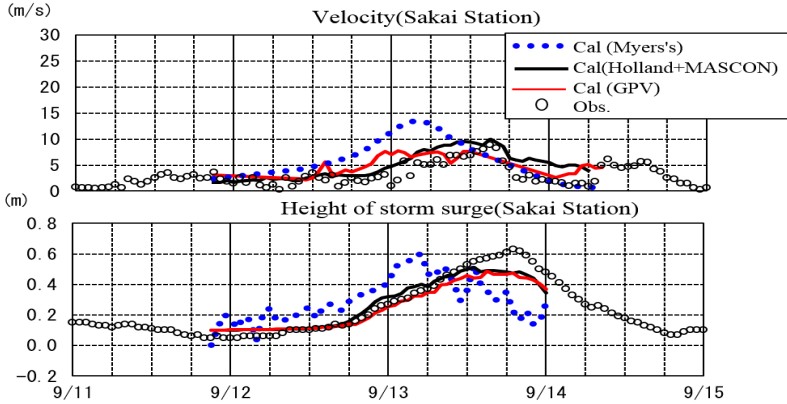

**Figure 15.** The results of storm surge based on typhoon models (Typhoon No. 14 in 2003).

Figure 16 shows the tide levels and storm surge heights that were estimated for Typhoon No. 16 "Chaba" in 2004 using the Holland + Mascon model and GPV data at the Saigou and Sakai points. The modeling outcomes show that the reproduced results obtained by the analysis method proposed in this study are more accurate than those obtained using the GPV data. Therefore, the reproducibility of the proposed model was evaluated to be very high.

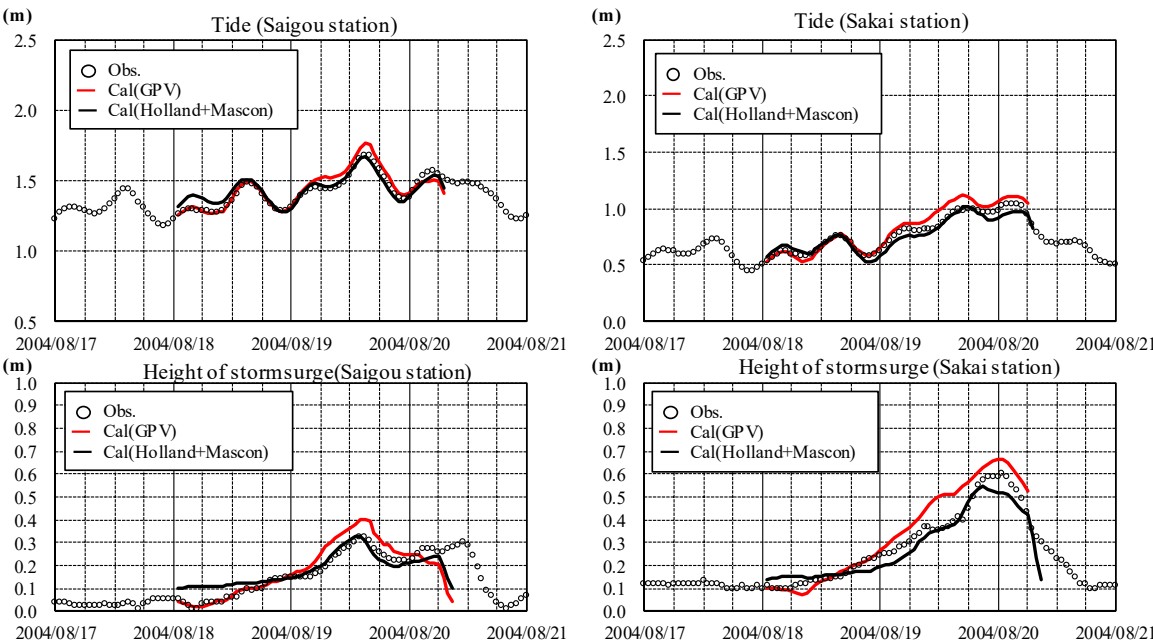

**Figure 16.** Verification results of the numerical simulation of the tide level and storm surge height.

## 5. Discussion

To develop a parametric model considering wind and pressure fields, which vary with respect to topographical features, and overcome the limitations of the Myers concentric circle model, this study analyzed the occurrence characteristics of storm surges through a test bed and reproduced the storm surge heights according to this parametric model. The primary results of this study are summarized below.

In total, 460 previous typhoons for the period 1951–2016 were classified based on four tracks on the test bed in order to analyze their regional characteristics. As a result, a maximum storm surge height of 0.630 m, with an occurrence frequency of 29%, was recorded for the typhoons passing along Track 1 between South Korea and Japan (Table 2). Furthermore, this study identified regional characteristics; it was found that the maximum tidal level occurred 10–30 h after the central pressure of the typhoon observed at the nearby observatory (Sakai Observatory) was the lowest among three previous typhoons with recorded maximum storm surge heights (along Track 1) (Figure 3). This observation indicates that the maximum storm surge height occurs after the typhoon passes through this region. Therefore, this phenomenon must be reproduced in the numerical model.

When a typhoon develops in the Pacific Ocean without being subjected to the topographical influences of surrounding regions, the shape of such a typhoon is similar to a concentric circle (Figure 8a). However, as the typhoon moves toward the north, its wind field is altered due to topographical influences of the surroundings. Consequently, the shape of the typhoon collapses (Figure 8b). Therefore, wind fields can be estimated using the Myers concentric circle model for regions neighboring the Pacific Ocean. However, the limitations of this concentric circle model become evident when calculating the wind fields in regions experiencing significant topographical influences, such as the East Sea.

Analyses of the observation data explain why the maximum abnormal tidal level occurs 10–30 h after the passage of a typhoon. When the wind field of a typhoon passes between South Korea and Japan, the typhoon loses its concentric-circle shape due to the topographical influences of the surroundings (Figure 7b). The resulting impacts on the wind field cause a delay in the occurrence of the maximum abnormal tidal level. Thus, a typhoon model accounting for the topographical influences of the surroundings is essential for calculating the wind field of a typhoon. In particular, it is necessary to calculate the wind field using a parametric model in order to simulate a virtual typhoon (the largest typhoon).

In this study, grid point value (GPV) data were used to analyze the location of central pressure and the radius of the pressure distribution for a typhoon moving northward (Figure 9). As the typhoon passes the 40° north latitude, the radius of pressure distribution (995 hPa; the pressure at the point where the wind speed is 17 m/s and classified as a typhoon) increases, and the effective range of the typhoon also increases. Therefore, the shape of the typhoon changes rapidly from a concentric circle to an elliptical shape owing to the topographic influence of surrounding regions. Subsequently, the typhoon moves northward toward the East Sea. Therefore, calculating the wind field in this region requires a typhoon model that considers regional characteristics, rather than a concentric circle model.

This study employed a combination of the Holland and Mascon models in order to estimate the wind fields caused by topographical influences, based on the diffraction states of the wind field. The results indicate that, although the maximum peak tidal level in the test bed is slightly underestimated with respect to the region, the phase depending on the time series is more consistent compared to that when using GPV data (Figure 16).

Numerical calculations considering the effects of waves as well as the linear and nonlinear interaction effects of tides on storm surge estimations can be well reproduced by using various analysis techniques. Approaches to predict information regarding typhoons after a maximum of 36 h, while considering extreme wave effects as well as the influence of wind based on meteorological data, have previously been employed in disaster prevention systems. On the contrary, this study aimed to evaluate topographical influences by considering the maximum typhoon intensity, in terms of wind speed and central pressure according to the manual [3], among previous typhoons by means of a test bed. Consequently, a parametric typhoon model accounting for wind and pressure field effects was achieved.

## 6. Conclusions and Future Works

To predict the largest-class typhoon occurrences, a parametric typhoon model that can interpret its wind field is required. To produce a past typhoon simulation, the model's reproducibility is best when GPV data are used, rather than a wind field analysis by typhoon model. However, since an artificial wind field analysis is impossible, a parametric typhoon model that can analyze the wind field distribution changes according to regional topographic features is important.

For the storm surge height estimation of the target region, the prediction accuracy of pressure and wind fields of the typhoon moving north toward the East Sea was critical. Moreover, the reproducibility of pressure and wind fields using the Myers model assuming concentric circles was poor because the typhoon shape was distorted from a concentric circle to an oval owing to the geographical characteristics of Korea and Japan. In the study of storm surge height estimation, the best way to calculate the surge height is to use typhoon wind field and pressure GPV data, which has a very high reproducibility. However, although GPV data present an advantage in disaster prevention system applications by providing predicted typhoon information based on the past and up to 36 h in advance of occurrence, they are not suitable for estimating storm surge height because the main typhoon parameters (central pressure and maximum wind speed) cannot be arbitrarily set. Therefore, a parametric model is required. Hence, a typhoon model that can consider topographical effects is required. The new typhoon model that considered the characteristics of typhoons correctly reproduced the phenomenon where the maximum sea level appeared after the typhoon passed through the target region. Further research will be conducted regarding the construction of a flood hazard map to estimate the storm surge height caused by large typhoons using the new typhoon model, which can consider geographical characteristics.

**Author Contributions:** Conceptualization, Y.-j.K. and J.-s.Y.; methodology, Y.-j.K.; software, Y.-j.K.; validation, Y.-j.K.; formal analysis, Y.-j.K.; data curation, Y.-j.K. and T.-w.K.; visualization, writing–original draft preparation, Y.-j.K.; writing–review and editing, Y.-j.K.; supervision and project administration, J.-s.Y.; All authors have read and agreed to the published version of the manuscript.

**Funding:** This research was a part of the project titled "Practical Technologies for Coastal Erosion Control and Countermeasure", funded by the Ministry of Oceans and Fisheries, Korea. (No. 20180404).

**Acknowledgments:** We would like to thank the anonymous reviewers and the editor for their efforts to improve the manuscript.

**Conflicts of Interest:** The authors declare no conflict of interest.

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
