# Peer review of "Study on Storm Surge Using Parametric Model with Geographical Characteristics"

_water, doi:10.3390/w12082251_

Round 1
Reviewer 1 Report
As stated when the manuscript was submitted for the first time the work is interesting, and the subject is worthy of research.
I find that authors introduced number of changes according to all three reviews. Now the manuscript looks much better thought-of, well written and finally well balanced between methodology, results and conclusions. However, I still find that the execution of the article requires some improvements to proceed with its publication in the journal in my opinion. For this I advise a minor revision.
In my opinion although authors added several references on storm surge studies the manuscript still miss some of recent articles that deals with the subject of modelling of this phenomena.
Finally I still miss a discussion part. Authors jumps form the result section right to conclusions. Please present a in-depth discussion comparing your method and results with existing, published articles.
Author Response
Thank you for your comments. I checked and modified my paper under your comments.
- Adding the literature review (line 51~69)
- Adding the discussion part (line 240~292)

Reviewer 2 Report
- The authors “do not” well address all my comments for the previous manuscript version. The responses for reviewers have to be in a separate document rather than just type “in totality modified with your comment”.
- Additionally, some references relative to the effect of typhoon winds on storm surge and wave simulation should be considered in your study (e.g., Shih-Chun Hsiao, et.al., 2020. Numerical Simulation of Large Wave Heights from Super Typhoon Nepartak (2016) in the Eastern Waters of Taiwan. J. Mar. Sci. Eng., 8, 217; Shih-Chun Hsiao, et.al., 2019. Quantifying the contribution of nonlinear interactions to storm tide simulations during a super typhoon event. Ocean Engineering, 194, 106661; Wei-Bo Chen, et. al., 2019. Wind forcing effect on hindcasting of typhoon-driven extreme waves. Ocean Engineering, 188, 106260).
Author Response

(The authors gave the same response as above.)

Round 2
Reviewer 2 Report
The authors have well addressed all my comments point by point. The manuscript has been improved significantly. I suggest this paper can be accepted for publication in the Water journal in the present form.
Author Response
Thank you for your comments.
This manuscript is a resubmission of an earlier submission. The following is a list of the peer review reports and author responses from that submission.
Round 1
Reviewer 1 Report
In my opinion the authors succeed in improving the existing methods for typhoon prediction and modelling in the Korea-Japan region. The combination of Myers and Holland models seems to fit the observed wind an storm surge variables much better than the Myers model alone. Although the fit is still not perfect, this work means a substantial advance in the correct prevision of typhoons in the area.
I recommend publication, after just some minor corrections. Comments and suggestions are included in the file attached.

Reviewer 2 Report
The article is interesting, and the subject is worthy of research. However, the execution of the article and the research as well as presentation itself requires some important improvements to proceed with its publication in the journal in my opinion. For this I advise to reject the paper and encourage authors to resubmit after introducing significant changes to the manuscript.
In a number of cases, you had facts, information, ideas or methods that were not your own, which had no in-text citation. It is very important to stick a reference at the end of the paragraph. See for example lines: 41-44, 51-53, etc.
It is unclear to me what is the significance and innovation of the work discussed in the paper, compared to other similar works. Authors state that in the manuscript they propose a new analysis method to allow for topographic consideration when calculating wind and pressure fields. Please add information on how this problem been solved by different authors. Or state clearly that this is the first time the land topography is considered in this type of modelling. Literature review in this subject is necessary.
In introduction authors miss number of recent articles that deals with storm surge modelling that suggests that the literature review is still to be done. Citing only nine literature positions in the “Introduction” (actually only 5 , as 4 are self-citation) looks really poor and one more time suggests that the literature review has to be redone.
I miss a general area representation. I advise to add one general figure representing the described area within a bigger perspective as not all international readers might be familiar with the study area or city names only. Additionally all figures with maps are required to have a basic map attributes like north arrow and scale.
In lines 133-142 authors try to prove that the typhoon shape was altered by the geographical characteristics. This is a crucial part as this part explains why they later implement the geographical characteristic to the model. Unfortunately in my opinion the explanation in this paragraph is very poor. Stating some fact and presenting conclusion is no enough. Please prove clearly this part. Please support your findings with literature.
The manuscript details several techniques used in different models, but the novelty of the paper is not clear. It is a summary of already used models combined together while I cannot see new insights regarding the application, processing, and interpretations. The authors specify very vast aims, which are too general and basic for a scientific paper.
The structure of the paper have to be improved. The manuscript need a better conceptual framework section. Now it is hard to distinguish which part is methodology and which results. Actually I miss a discussion part. Seems this section had been omitted totally and authors just jumped to conclusions. This again goes back to the Introduction and lack of earlier literature review.
Reviewer 3 Report
This work focuses on simulating storm surge suing a parametric model with geographical characteristics. Although the content is within the scope of the Water journal, the description of methodology should be improved. My suggestion is a major revision. Major Comments: 1. The description of the methodology for this study is too vague. For example, how did the authors combine the Myers model and the Holland model or the Mascon model? What is the grid point value? How does the grid point value apply in the typhoon model? More details about the methodology should be provided in the manuscript. 2. What kind of wind field does the author use for storm surge simulation? Myers+ Holland or Myers+ Mascon or Holland+ Mascon? More details should be provided in the manuscript. 3. What is the best combination of the typhoon wind field for storm surge simulation? 4. Only a typhoon event (Typhoon Maemi) was used for storm surge simulation in this study, however, in terms of model validation, 2 or 3 events are needed. Minor Comments: 1. Figure 2 was lost in the manuscript. 2. Line 95, please remove “of” in the caption of Table 2. 3. What does “case” mean in Table 2? 4. The caption of Table 2 has to revise as “The maximum tide and surge of the typhoon from the different track”. 5. Figure 5 is not a planar distribution of the wind direction and velocity. 6. Line 143, which typhoon model was used? (a)(b)(c)(d) should be marked in Figure 6. 7. Lin 212, which typhoon model was used?
